# Arginine Decarboxylase Is Essential for Pneumococcal Stress Responses

**DOI:** 10.3390/pathogens10030286

**Published:** 2021-03-02

**Authors:** Mary Frances Nakamya, Moses B. Ayoola, Leslie A. Shack, Mirghani Mohamed, Edwin Swiatlo, Bindu Nanduri

**Affiliations:** 1Department of comparative Biomedical Sciences, College of Veterinary Medicine, Mississippi State, MS 39762, USA; mfn35@msstate.edu (M.F.N.); mba185@msstate.edu (M.B.A.); shack@cvm.msstate.edu (L.A.S.); 2Department of Molecular and Cell Biology, College of Liberal Arts and Sciences, University of Connecticut, Storrs, CT 06269, USA; mirghanimohamed274@gmail.com; 3Section of Infectious Diseases, Southeast Louisiana Veterans Health Care System, New Orleans, LA 70112, USA; edwin.swiatlo@va.gov

**Keywords:** *Streptococcus pneumoniae*, polyamines, oxidative stress, nitrosative stress, acid stress, arginine decarboxylase

## Abstract

Polyamines such as putrescine, cadaverine, and spermidine are small cationic molecules that play significant roles in cellular processes, including bacterial stress responses and host–pathogen interactions. *Streptococcus pneumoniae* is an opportunistic human pathogen, which causes several diseases that account for significant morbidity and mortality worldwide. As it transits through different host niches, *S. pneumoniae* is exposed to and must adapt to different types of stress in the host microenvironment. We earlier reported that *S. pneumoniae* TIGR4, which harbors an isogenic deletion of an arginine decarboxylase (Δ*speA*), an enzyme that catalyzes the synthesis of agmatine in the polyamine synthesis pathway, has a reduced capsule. Here, we report the impact of arginine decarboxylase deletion on pneumococcal stress responses. Our results show that *ΔspeA* is more susceptible to oxidative, nitrosative, and acid stress compared to the wild-type strain. Gene expression analysis by qRT-PCR indicates that thiol peroxidase, a scavenger of reactive oxygen species and *aguA* from the arginine deiminase system, could be important for peroxide stress responses in a polyamine-dependent manner. Our results also show that *speA* is essential for endogenous hydrogen peroxide and glutathione production in *S. pneumoniae*. Taken together, our findings demonstrate the critical role of arginine decarboxylase in pneumococcal stress responses that could impact adaptation and survival in the host.

## 1. Introduction

Acidification of the phagolysosome and production of reactive oxygen species (ROS)/reactive nitrogen species (RNS) by immune cells are major defense mechanisms against invading pathogens [1,2]. ROS are produced by specialized phagocytic cells (macrophages and neutrophils) through the nicotinamide adenine dinucleotide phosphate oxidase (NADPH) complex upon detection of pathogen-associated molecular patterns. RNS are generated by inducible nitric oxide synthases (iNOS), which generate NO^•^, contributing to pathogen killing [3,4]. Phagolysosome maturation generates an acidified environment that reduces pH from 6.5 to 4.0 and is a key defense mechanism against invading pathogens [1]. Thus, to colonize, inhabit, and cause disease in the host, pathogens need to circumvent the antimicrobial effects of the oxidative, nitrosative, and acidified milieu in the host [5]. Bacterial protective measures against ROS and RNS include detoxifying enzymes such as catalases, peroxidases, superoxide dismutase, and DNA repair systems [6]. Bacterial defense against acid stress involves the agmatine deiminase system, ATPase system, and *ciaRH,* a two-component regulatory system. Polyamines that are ubiquitous polycationic molecules at physiological pH regulate many bacterial processes, including oxidative and acid stress responses [7,8,9,10]. Polyamines protect bacteria against the toxic effects of ROS by directly scavenging free radicals, binding and stabilizing nucleic acids, or inducing operons that encode enzymes capable of inactivating oxidants [8,11]. Our earlier studies showed that polyamine biosynthesis and transport are vital for pneumococcal survival in murine models of colonization, pneumonia, and sepsis [12]. Recently, we reported that deletion of *speA*, a gene that encodes an arginine decarboxylase and a polyamine transporter (*potABCD*), resulted in reduced intracellular polyamine concentrations of putrescine, spermidine, and cadaverine and an unencapsulated phenotype [13,14,15]. Characterization of *ΔspeA* using proteomics and transcriptomics [13,14,15] indicated an increased carbon flow through the pentose phosphate pathway (PPP). Upregulation of the PPP constitutes a cellular response to oxidative stress by maintaining the nicotinamide adenine dinucleotide (NADH)/NADPH redox balance and generating ribose-5-phosphate to increase nucleotide synthesis to combat ROS-mediated DNA damage [16]. 

Gram-positive pathogens such as *Streptococcus pneumoniae* (pneumococcus) cause significant morbidity and mortality worldwide [17]. After colonizing the human nasopharynx asymptomatically, pneumococci can, under specific conditions, translocate to the middle ear, lung, blood, and brain and cause both invasive and non-invasive diseases. In these environments, the pneumococcus encounters various stress conditions [18]. Since polyamines are known to be involved in bacterial stress responses, impaired polyamine synthesis in Δ*speA* is expected to adversely affect pneumococcal stress responses that are critical for in vivo fitness. To validate the oxidative signature depicted by the increased PPP in Δ*speA* in our earlier studies, in this study, we determined the susceptibility of Δ*speA* to oxidative, nitrosative, acid, and thermal stress in vitro. Our results show that Δ*speA* is more susceptible to hydrogen peroxide, superoxide (potassium tellurite), nitrosative (*S*-nitrosoglutathione), and acid stress compared to the wild-type (WT) strain. However, deletion of *speA* did not have a noticeable impact on pneumococcal survival during thermal stress. We also determined intracellular pH, endogenous hydrogen peroxide, and glutathione production (reduced/oxidized glutathione (GSH/GSSG) ratio) in the WT and Δ*speA* strains. Our results show that pneumococcal hydrogen peroxide and glutathione production are dependent on *speA* but not intracellular pH. Furthermore, to identify specific pneumococcal hydrogen peroxide resistance mechanisms, we measured the expression of genes known to be involved in oxidative stress responses using qRT-PCR. Our results show that the enhanced susceptibility of Δ*speA* to exogenous hydrogen peroxide could be due to *speA* deletion itself, which altered the expression of genes involved in polyamine biosynthesis, protein repair, and detoxification. This study demonstrates that deficiency of *speA* impairs the pneumococcus’s stress responses that could impact its in vivo survival.

## 2. Results

### 2.1. SpeA-Deficient Pneumococci Are Susceptible to Oxidative Stress

Our results show that compared to the WT strain, *speA* is more susceptible to hydrogen peroxide stress. When cultured in the presence of low concentrations of exogenous hydrogen peroxide (0.5, 0.75, and 1 mM), there was no significant effect on the growth of WT and Δ*speA* strains at 15 and 30 min post-exposure (data not shown). However, in the presence of 2.5 or 5 mM H_2_O_2_, survival of Δ*speA* reduced by 33% and 66% relative to the WT strain (0% and 1.1%) at 15 min post-exposure, respectively (Figure 1A). At 30 min post-exposure, Δ*speA* viability significantly reduced by 92.4% and 99.5% compared to the WT strain 12% and 29%, at 2.5 and 5 mM H_2_O_2_, concentrations, respectively (Figure 1B). Susceptibility of Δ*speA* complemented with the pABG5-*speA* construct was comparable to that of the WT strain at both time points and H_2_O_2_ concentrations (Figure 1A,B).

We measured the impact of the deletion of *speA* on H_2_O_2_ production. Our results show that Δ*speA* generates less H_2_O_2_ (40%; *p* = 0.0004) compared to the WT strain (data not shown). This result indicates that it is not higher endogenous H_2_O_2_ production that renders Δ*speA* more susceptible to exogenous H_2_O_2_.

Like hydrogen peroxide, Δ*speA* was more susceptible to superoxide stress generated by potassium tellurite. There was no significant difference in the viability of Δ*speA* and WT strains at 0.1 mM and 15 min post-exposure, but a significant reduction in Δ*speA* viability (~27%) was observed at a concentration of 0.2 mM potassium tellurite (Figure 2A). At 30 min post-exposure, Δ*speA* viability significantly reduced by ~23% and ~73% compared to the WT strain (0% and ~27%) at 0.1 mM and 0.2 mM potassium tellurite concentrations, respectively (Figure 2B). Susceptibility of the complement strain to potassium tellurite was comparable to that of the WT strain at both time points and concentrations (Figure 2A,B).

### 2.2. Impact of SpeA Deletion on Pneumococcal Gene Expression during Hydrogen Peroxide Stress

To identify gene expression changes that could explain the higher susceptibility of the *speA*-deficient strain to hydrogen peroxide, we measured the expression of genes known to be part of the pneumococcal oxidative stress response using qRT-PCR. Significant changes in gene expression (*p* ≤ 0.05 and fold change ≥ 2) in *ΔspeA* relative to untreated and hydrogen-peroxide-treated WT strains were compared to distinguish the impact of gene deletion itself (untreated) from the *ΔspeA*-dependent response to H_2_O_2_ (Table 1 and Appendix A). Deletion of *speA* resulted in reduced expression*,* of high-temperature requirement A (*htrA*) sensor histidine kinase (*ciaH*), NADH oxidase (*noxA*), and genes involved in polyamine biosynthesis but led to increased expression of serine protease (*prtA)* and a gene that encodes the polyamine ATP-binding cassette transporters (ABC) transporter substrate-binding protein (*potD*) (Table 1). Exposure of Δ*speA* to H_2_O_2_ significantly downregulated thiol peroxidase (*tpxD)* and agmatine deiminase (*aguA*) and upregulated the expression of *noxA,* genes that encode a siderophore transport protein (*fhuD*), and a manganese ABC transporter substrate-binding lipoprotein (*psaA*) (Table 1). Altered expression of these genes in *ΔspeA* shows that arginine decarboxylase is important in pneumococcal hydrogen peroxide stress responses. Surprisingly, most key genes known to be involved in pneumococcal oxidative stress responses were either not significantly altered or had <twofold change in expression (Appendix A).

We observed a significant change in the expression of genes involved in glutathione (GSH) metabolism that were below our fold-change cutoff (Appendix A). Since GSH metabolism is known to be involved in oxidative stress responses, we measured the ratio of reduced/oxidized glutathione (GSH/GSSG) in the mutant and WT strains. We observed a significantly lower GSH/GSSG ratio (0.37 ± 0.09) in the mutant strain compared to the WT strain (1.30 ± 0.06, *p* < 0.0001). A GSH/GSSG ratio of less than 1 indicates reduced glutathione production, which suggests an impaired redox balance in the mutant strain. These results clearly show that deficiency of polyamines affects the pneumococcal transcriptome, which impairs the redox balance and renders *ΔspeA* more susceptible to stress.

### 2.3. SpeA Is Required for Pneumococcal Nitrosative Stress Responses

Compared to the WT strain, Δ*speA* was significantly more susceptible to *S*-nitrosoglutathione (*GSNO*), a nitric oxide producer. Exposure to 2.5 mM *GSNO* resulted in a significant reduction in the percentage survival of Δ*speA*, which ranged between 37% at 15 min and 74% at 60 min post-exposure compared to the WT strain (0% at 15 min and 36% at 60 min post-exposure). There was no significant difference in the survival of the complement pABG5-*speA* strain and the WT strain in the presence of *GSNO* (Figure 3).

### 2.4. Effect of Selection of SpeA on Pneumococcal pH_i_

We hypothesized that increased susceptibility of the mutant strain to oxidative and nitrosative stress could be due to an intracellular acidified environment due to the deletion of *speA* and measured intracellular pH (pH_i_). Our result shows that pH_i_ of Δ*speA* (~7.3 ± 0.01) differed from that of the WT strain (~7.5 ± 0.01) by 0.2 units (*p* < 0.0001; Figure 4). The marginal change in pH_i_ in ∆*speA* is within the physiological range, suggesting that susceptibility to oxidative and nitrosative stress is independent of pH_i_.

### 2.5. Effect of Deletion of SpeA on Pneumococcal Acid and Thermal Stress Responses

We observed that Δ*speA* is more susceptible to acid and not thermal stress. Comparison of WT, Δ*speA*, and pABG5-*speA* strains in Todd–Hewitt broth with 0.5% yeast extract (THY) at different pH values showed inhibition of growth of all strains at pH ≤ 5.5 (data shown). Comparison of the growth rate and maximum optical density of the WT, Δ*speA*, and pABG5-*speA* strains at pH 6.0 and 7.4 identified no significant difference at *p* ≤ 0.05. However, there was a significant difference between the lag phase of the mutant strain compared to the WT/pABG5-*speA* strain at pH 5.7 (*p* ≤ 0.01) (Figure 5A). Δ*speA* had no noticeable growth at pH 5.7, and normal growth for Δ*speA* was observed only at pH 6.0 and above (Figure 5B). Pneumococci must adapt to varying temperatures at different sites in the human body. Growth of WT and Δ*speA* strains was comparable at 30 °C and 40 °C (Appendix A), indicating that *speA* function may not be necessary for pneumococcal thermal stress responses.

## 3. Discussion

The major finding in this study is that *speA* is important for pneumococcal oxidative, nitrosative, and acid but not thermal stress responses. Our results also show that glutathione and H_2_O_2_ production is dependent on *speA* in pneumococci, while the reverse is true for pH_i_. Our qRT-PCR results show that enhanced susceptibility of Δ*speA* to exogenous hydrogen peroxide could be due to the deletion of *speA* itself and its impact on the expression of other genes involved in manganese and iron transport, polyamine biosynthesis, protein repair, and detoxification (Table 1). Polyamines are known to be involved in ROS scavenging, pH homeostasis, and regulation of antioxidant systems. Therefore, reduced polyamine synthesis that results in altered levels of intracellular polyamines is expected to render *ΔspeA* more susceptible to stress [4,8,9,19,20,21,22,23,24,25,26,27]. Indeed, when we characterized the proteome and transcriptome of *ΔspeA*, we observed a shift in the carbon flow toward the pentose phosphate pathway [13,14], a signature of oxidative stress [16]. In this study, measurement of parameters indicative of cellular stress and rigorous analysis of the susceptibility of WT and *ΔspeA* strains to various stressors validated the previously reported signature of oxidative stress at the molecular level.

Although catalase negative, *S. pneumoniae* generates high levels of H_2_O_2_ (2 mM) via pyruvate oxidase [28,29]. Intrinsic H_2_O_2_ generation is critical for pneumococcal pathogenesis. H_2_O_2_ can inhibit or kill other common inhabitants of the respiratory tract. It is cytotoxic to host cells, causes apoptosis in respiratory epithelial cells, and promotes colonization of the upper respiratory tract [30,31,32]. Therefore, reduced ability to generate H_2_O_2_ in *ΔspeA* observed in this study could reduce the colonization and invasive potential of *ΔspeA*. Increased sensitivity of Δ*speA* to exogenous H_2_O_2_ and superoxide radicals reported here suggests a role of polyamines in pneumococcal oxidative stress responses, which is consistent with earlier reports. Putrescine and spermidine protect cells from ROS by increasing the expression of regulators or genes that encode free-radical scavengers [8,24,25,33]. Spermine and spermidine are known to scavenge ROS and work synergistically with superoxide dismutase to reduce single-stranded DNA breakages due to ROS [25,34,35]. In addition, cadaverine protects *Escherichia coli* and *Vibrio vulnificus* in high-oxygen environments and superoxide stress, respectively [20,36]. We recently reported that *speA* deletion resulted in significant reduction in the intracellular levels of agmatine, the product of arginine decarboxylase activity of SpeA. Therefore, this study implicates that agmatine is a critical polyamine that could regulate pneumococcal stress responses, which warrants further investigation in the future. However, superoxide susceptibility results of this study contradict our earlier report [12] where we observed similar survival rates for WT and ∆s*peA* strains when treated with superoxide produced by paraquat. This discrepancy could be due to the difference in the compounds used to generate superoxide anions and/or the higher concentration of paraquat used. The high concentration of paraquat that we used earlier could have been equally toxic to both WT and mutant strains.

We carried out qRT-PCR with RNA-isolated pneumococci exposed to 2.5 mM H_2_O_2_ for 15 min. The 15 min exposure time, which is shorter than the pneumococcal-doubling time, represents transcriptional changes due to the deficiency of *speA* during H_2_O_2_ stress, without the confounding effects of growth and adaptation. These results show that *speA* affects the expression of genes involved in pneumococcal stress responses, which concurs with reports that polyamines regulate the expression of genes involved in oxidative stress responses [33]. Pneumococci deficient in *tpxD* and *htrA* are susceptible to exogenous H_2_O_2_, and Δ*htrA* was attenuated in murine models of nasopharyngeal colonization, pneumonia, and bacteremia [37,38,39,40,41]. Therefore, reduced expression of *tpxD* and *htrA* in Δ*speA* could result in reduced survival in the host, which could explain its reported in vivo attenuation [37]. Furthermore, a reduced GSH/GSSG ratio indicates impaired redox homeostasis in Δ*speA*, which could further render the mutant susceptible to oxidative stress. Reduced glutathione and glutathione metabolism are involved in oxidative stress responses in other bacterial pathogens [42,43,44,45,46]. Controlled influx and efflux of cellular Mn is important for normal growth and oxidative stress responses in many pathogenic bacteria [47,48]. The Fenton reaction with H_2_O_2_ generates hydroxyl radicals, which is the primary cause of damage to biomolecules such as DNA. Increased expression of the iron transporter *fhuD* and the manganese transporter *psaA* could exacerbate the effects of stress in the mutant, which could impair pneumococcal survival in vivo, especially during transition between different niches. Our results agree with the reported role of polyamines in the regulation of bacterial stress responses by modifying the expression of stress response genes [33].

Δ*speA* was more sensitive to nitrosative stress compared to the WT strain. Findings of this study agree with reports on the protective role of polyamines in nitrosative stress responses: putrescine and spermidine in *Salmonella typhimurium* against nitrosative stress [22] and cadaverine against nitrosative stress in *E. coli* [49]. The increased susceptibility of the mutant to nitrosative stress will impact pneumococcal in vivo adaptation in host immune cells such as macrophages, epithelial cells, and dendritic cells known to produce nitric oxide via inducible nitric oxide synthases for bacterial clearance [50]. We hypothesized that the increased sensitivity of Δ*speA* to oxidative and nitrosative stress could be due to impaired buffering capacity. However, the pH_i_ of WT and Δ*speA* strains remained within the physiological pH range (Figure 4). These results conform to reports that bacteria maintain their pH_i_ within a narrow pH range despite exposure and growth under varied extracellular pH conditions [51,52]. Although pneumococci are reported to be tolerant of pH 4.4 [51] in phagosome vesicles, our results show increased sensitivity of Δ*speA* to acid stress, suggesting that impaired polyamine synthesis renders pneumococci more susceptible to acid stress. During the lag phase, bacteria synthesize proteins and other molecules necessary for replication. A prolonged lag phase in the mutant at all pH values suggests that *speA*’s indirect effects could be essential for replication. These results concur with our earlier study where we reported reduced expression of genes involved in protein and capsule synthesis in Δ*speA* [13,14]. These results are consistent with studies on other pathogenic bacteria that report the neutralizing effects of polyamines (cadaverine) at low pH [26,36,53,54]. Since the synthesis of polyamines via the decarboxylation of amino acids consumes a proton and generates ammonia, which protects bacteria against acid stress [54,55,56], *speA* deficiency could render the mutant more susceptible to low pH. Nevertheless, findings of this study contradict our earlier conclusion [12]. Compared to our earlier report, in this study, we characterized the growth of the strains over a wide range of pH (4.0–7.8), while the earlier study used only pH 5.5, which showed no significant difference in the survival of the WT strain. Results at pH 5.5 are similar with the findings of this study. Growth kinetics of WT and *ΔspeA* strains at different temperatures indicated that deletion of *speA* has no noticeable impact on pneumococcal thermal stress responses (Supp. Appendix A). In *S. pneumoniae* D39 [7], the substrate-binding protein *potD* was reported to be essential for thermal stress responses. We recently reported that *speA* deletion resulted in significant reduction in the intracellular levels of only agmatine, the product of arginine decarboxylase. Therefore, this study indicates that agmatine is a critical polyamine that could regulate pneumococcal stress responses, which warrants further investigation in the future. Organisms alter their gene expression to limit energy-consuming processes. Inhibition of capsular polysaccharide (CPS) synthesis in Δ*speA* by metabolic reprogramming could be an adaptive response to counter increased intracellular oxidative and nitrosative stress coupled with impaired stress responses. This is supported by the downregulation of genes involved in carbohydrate metabolism and nucleotide synthesis in favor of the pentose phosphate pathway that generates NADPH, a cofactor for antioxidant systems [13,14].

In summary, deletion of arginine decarboxylase, an enzyme from the putrescine/spermidine biosynthesis pathway, adversely effects pneumococcal oxidative, nitrosative, and acid stress responses, which impacts its ability to survive in the human host. Expression of pneumococcal oxidative stress response genes such as *tpxD, htrA*, and *aguA* is dependent on *speA*, which warrants further investigation. For the first time, our data suggest that *speA* regulates the production of hydrogen peroxide and glutathione in pneumococci. These results once again illustrate the need for polyamine homeostasis in bacterial pathogens, the disruption of which has implications for physiology and virulence, providing insight into the potential for identifying novel therapeutic targets. However, comprehensive omics studies are needed to elucidate a crosstalk between polyamine metabolism, pneumococcal stress responses, capsule synthesis, and pathogenesis.

## 4. Methods

### 4.1. Bacterial Strains and Growth Conditions

*S. pneumoniae* serotype 4 clinical isolate TIGR4 [57], Δ*speA*, and pABG5-*speA* [14] strains were used in this study [14]. Pneumococci were grown in Todd–Hewitt broth with 0.5% yeast extract (THY) or on blood agar plates (BAP) at 37 °C in 5% CO_2_ unless otherwise specified. For all stress susceptibility assays, the percentage survival of cells treated with a chemical stressor was calculated relative to untreated cultures. All assays were performed in triplicate in three independent experiments.

### 4.2. Hydrogen Peroxide Survival Assay

Mid-log phase cultures of TIGR4, Δ*speA*, and pABG5-*speA* (optical density at 600 nm (OD_600nm_) = 0.4–0.5) were supplemented with final concentrations of hydrogen peroxide ranging from 0.5 to 5 mM and incubated at 37 °C with 5% CO_2_ for 30 min. The control reactions contained untreated bacteria. At 15 min intervals, aliquots were serially diluted in sterile phosphate-buffered saline (PBS) and plated on BAP for colony-forming unit (CFU) enumeration.

### 4.3. Potassium Tellurite Susceptibility Assay

Mid-log phase cultures of TIGR4, Δ*speA*, and pABG5-*speA* were centrifuged at 10,000× *g* for 2 min and cells suspended in PBS. Bacteria (10^8^ CFU/mL) in 1 mL of PBS were exposed to 0.1 and 0.2 mM potassium tellurite (Millipore-Sigma, St. Louis, MO, USA) and incubated for 30 min at 37 °C in 5% CO_2_. The control reactions contained untreated bacteria. At 15 min intervals, aliquots were serially diluted in sterile PBS and plated on BAP for CFU enumeration.

### 4.4. Hydrogen Peroxide Production Assay

H_2_O_2_ generated from mid-log phase cultures of (10 mL) TIGR4 and Δ*speA* was compared using a quantitative peroxide assay (Pierce, Thermo Fisher Scientific Waltham, MA, USA) following the manufacturer’s instructions. Briefly, 1 mL of bacterial culture (10^8^ CFU/mL) was centrifuged at 10,000× *g* at 4 °C for 2 min and the supernatant filtered with a 0.22 µM filter. The concentration of H_2_O_2_ was measured in the filtrate_._ The assay was performed in triplicate in three independent experiments.

### 4.5. S-Nitrosoglutathione Susceptibility Assay

Mid-log phase cultures of TIGR4, Δ*speA*, and pABG5-*speA* were centrifuged at 10,000× *g* for 2 min and cells suspended in PBS. Bacteria (10^7^ CFU/mL) in 100 µL of PBS were supplemented to a final concentration of 2.5 mM *S*-nitrosoglutathione (*GSNO*; Millipore-Sigma, St. Louis, MO, USA) and incubated at 37 °C in 5% CO_2_ for 60 min. Control reactions contained only bacteria in PBS. At 15 min intervals, aliquots were serially diluted in sterile PBS and plated on BAP for CFU enumeration. CFUs were determined by serial dilution and plating on BAP, and results were expressed as the percentage survival of treated bacteria relative to untreated controls.

### 4.6. Measurement of Intracellular pH (pH_i_)

The intracellular pH (pH_i_) was determined using the method described in [58] with slight modifications. Briefly, mid-log phase (OD600 nm 0.4) cultures of TIGR4 and Δ*speA* grown in THY (n = 3) were harvested, washed, and suspended in PBS. Cells (10^8^ CFU) in 1 mL of the culture were loaded with 5 mM 2*’*-7*’*-bis(carboxyethyl)-5(6)-carboxyfluorescein-acetoxymethyl (BCECF-AM) dye (Millipore-Sigma, St. Louis, MO, USA) and incubated for 30 min at 30 °C in the dark. Cells were pelleted, washed, and re-energized with 10 mM glucose in PBS. To obtain an in vivo calibration curve, measured specifically for each strain, 400 µL of energized cells were pelleted and suspended in potassium buffers ranging from pH 6.5 to 8.0. Nigericin (1 mM) (Thermo Fisher Scientific, Waltham, MA, USA) was added to the samples (to equilibrate the intracellular pH of the cells to the pH of the surrounding buffer) and incubated at 37 °C for 5 min. Fluorescence was measured by a Synergy plate reader (BioTEk, Winooski, VT, USA), and a calibration curve was obtained by plotting fluorescence against the pH of the buffers. To measure the pH of individual samples, 200 µL of loaded and energized cells were added to the wells of a 96-well plate in duplicate and fluorescence was detected using a plate reader for 5 min. Next, 10 µM of carbonyl cyanide 3-chlorophenylhydrazone (CCCP) was added to one well (to serve as a control), and fluorescence was measured for another 5 min. CCCP is a protonophore that can uncouple the proton motive force and cause a sudden decrease in intracellular pH (Millipore-Sigma, St. Louis, MO, USA). Nigericin (1 mM) was added to both CCCP-treated controls and the untreated sample to equilibrate the pH_i_ of the bacteria to the pH of the buffer, and fluorescence was measured for an additional 5 min. Fluorescence levels were calculated for each of the controls and sample, and pH_i_ was interpolated from the calibration curve. The assays were carried out in triplicate in three independent experiments.

### 4.7. In Vitro Growth under Acid and Thermal Stress

To compare the growth of TIGR4, Δ*speA*, and pABG5-*speA* at varying pH, the pH of THY was adjusted using either HCl or NaOH. Growth (10^5^ CFU/mL) of all strains in THY with pH adjusted between 4 and 7.5 with an interval of 0.2 units was monitored for 24 h by measuring the optical density at 600 nm (OD_600_) using a Cytation 5 multifunction plate reader (BioTek, Winooski, VT, USA). For thermal stress, growth of all strains in THY at 30 °C, 37 °C, and 40 °C was monitored for 24 h by measuring OD_600_ using a Cytation 5 multifunction plate reader.

### 4.8. RNA Extraction and Quantitative Real-Time PCR

Quantitative reverse transcription–PCR (qRT-PCR) was used to compare gene expression changes in Δ*speA* in response to H_2_O_2_ stress compared to the WT strain (see Appendix A for a list of primers used in this study). To ensure amplification of a single specific product, primers were validated using melt-curve analysis with SYBR Green (Thermo Fisher Scientific Waltham, MA, USA). Total RNA was purified from cells at the mid-log phase in THY (pH 7.4) and cells exposed to 2.5 mM hydrogen peroxide for 15 min. RNA was extracted and purified using a RNeasy Midi kit and QIAcube (Qiagen, Valencia, CA, USA). RNA from H_2_O_2_-treated and control bacteria was isolated from three independent cultures. Purified RNA (7.5 ng/reaction) was reverse-transcribed into complementary DNA (cDNA), and qRT-PCR was performed using the SuperScript III Platinum SYBR Green One-Step qRT-PCR Kit (Thermo Fisher Scientific, Waltham, MA, USA), as previously described [14]. Relative quantification of gene expression was determined using the Stratagene M × 3005P qPCR system (Agilent, Santa Clara, CA, USA). Expression of selected genes known to be involved in oxidative stress responses [40], polyamine biosynthesis, and transport was measured and normalized to the expression of *gyrB* and fold change determined by the comparative C_T_ method.

### 4.9. Measurement of Intracellular Glutathione

The ratio of reduced and oxidized intracellular glutathione concentrations in TIGR4 and Δ*speA* were determined using the GSH/GSSG-Glo™ Assay Kit (Promega, Madison, WI, USA). Mid-log 10 mL cultures, n = 3 were harvested at 5000× *g* for 10 min at 4 °C, suspended in PBS, and transferred to beadbeater tubes (MP Biomedicals, Irvine, CA, USA). Cell suspensions were lysed with a FastPrep-24™ Classic benchtop homogenizer (45 s, 6.5 m/s × 3) (MP Biomedicals, Irvine, CA, USA) and clarified by centrifugation at 6000× *g* for 5 min at 4 °C. The extracts were then processed according to the manufacturer’s instructions. Luminescence was measured with a Cytation™ 5 cell imaging multi-mode reader (BioTek, Winooski, VT, USA) and used to calculate glutathione concentrations. Protein concentrations of the extracts were determined with the Pierce BCA Protein Assay Kit (Thermo Fisher Scientific, Waltham, MA, USA) and used to normalize glutathione concentrations. GSH/GSSG (reduced/oxidized glutathione) ratios were calculated from the normalized glutathione concentrations according to the kit manufacturer’s instructions.

### 4.10. Statistical Analysis

Significant differences in susceptibility between WT and deletion strains to different stressors, GSH/GSSG ratio, changes in endogenous H_2_O_2_, and changes in gene expression measured by qRT-PCR were determined by Student’s *t*-test at a *p*-value of ≤ 0.05. Gene expression changes identified at *p* ≤ 0.05 and a fold change of 2 were considered for biological interpretation. GrowthRates [59], a software tool that uses the output of plate reader files to determine the growth rate in the exponential phase, lag phase, and maximal optical density (OD), was used to analyze growth curves of the bacterial strains used in this study.

## Figures and Tables

**Figure 1 pathogens-10-00286-f001:**
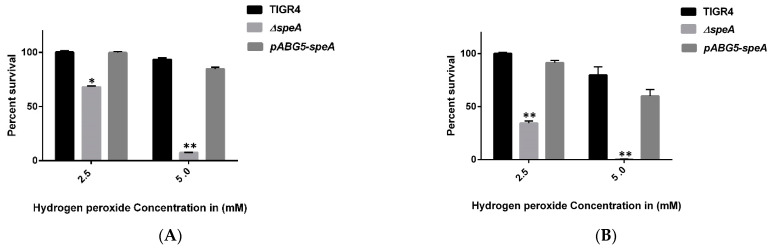
Hydrogen peroxide susceptibility of *Streptococcus pneumoniae* TIGR4, Δ*speA*, and pABG5-*speA*. Graph (**A**) shows bacterial sensitivity to 2.5 mM and 5.0 mM H_2_O_2_ at 15 min post-exposure. Graph (**B**) shows bacterial sensitivity to 2.5 mM and 5.0 mM H_2_O_2_ at 30 min post-exposure. The results represent an average of three independent experiments. The percentage survival relative to untreated controls is shown as a bar with the standard error of the mean, with * indicating *p* ≤ 0.05 and ** representing *p* ≤ 0.001, determined by Student’s *t*-test.

**Figure 2 pathogens-10-00286-f002:**
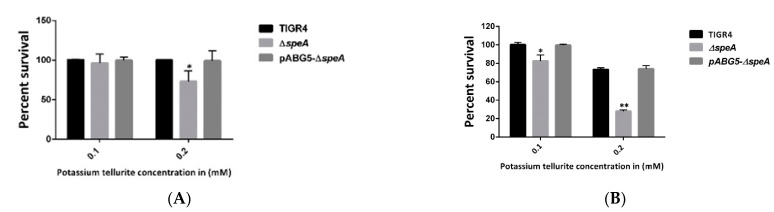
Potassium tellurite susceptibility of *S. pneumoniae* TIGR4, Δ*speA*, and pABG5-*speA*. Graph (**A**) shows bacterial sensitivity to 0.1 mM and 0.2 mM potassium tellurite at 15 min post-exposure. Graph (**B**) shows bacterial sensitivity to 0.1 mM and 0.2 mM potassium tellurite at 30 min post-exposure. The results represent an average of three independent experiments. The percentage survival relative to untreated controls is shown as a bar with the standard error of the mean, with * indicating *p* ≤ 0.05 and ** representing *p* ≤ 0.001, determined by Student’s *t*-test.

**Figure 3 pathogens-10-00286-f003:**
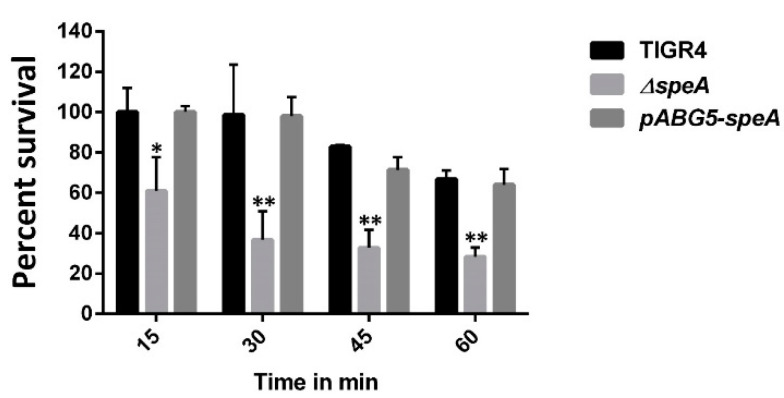
*S*-nitrosoglutathione susceptibility assay of *S. pneumoniae* TIGR4, *ΔspeA*, and pABG5-*speA*. The graph shows bacterial sensitivity to 2.5 mM GSNO at 15–60 min post-exposure. The results represent an average of three independent experiments. The percentage survival relative to untreated controls is shown as a bar with the standard error of the mean, with * indicating *p* ≤ 0.05 and ** representing *p* ≤ 0.001, determined by Student’s *t*-test.

**Figure 4 pathogens-10-00286-f004:**
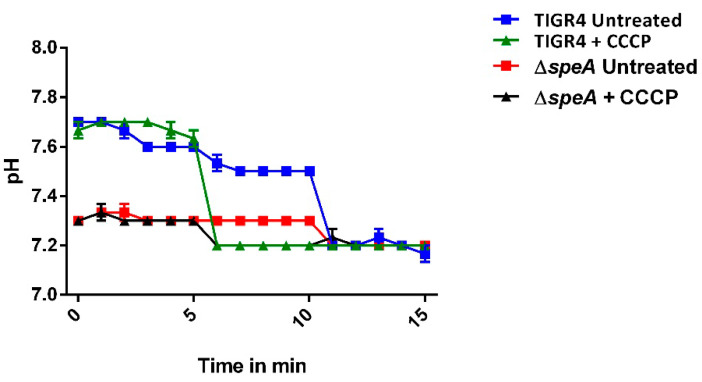
Intracellular pH of *S. pneumoniae* TIGR4 and Δ*speA*. Three replicates of TIGR4 and Δ*speA* were loaded with 5 mM of pH-sensitive fluorescence dye 2′-7′-bis(carboxyethyl)-5(6)-carboxyfluorescein - acetoxymethyl (BCECF-AM), washed with phosphate-buffered saline (PBS), and re-energized with 10% glucose, and baseline fluorescence readings were established in the first 5 min. Controls were supplemented with 10 µM carbonyl cyanide 3-chlorophenylhydrazone (CCCP) as a protonophore (triangles), and fluorescence of samples, including untreated ones (squares), was measured for an additional 5 min. The pH_i_ of untreated TIGR4 (blue squares) and Δ*speA* (red squares) was 7.5 and 7.3, respectively. Then, 20 µM of nigericin was added to both treated and untreated samples to dissipate transmembrane gradients over the last 5 min. Graphs represent the mean of three independent experiments. Statistical significance was determined by Student’s *t*-test at a significance level of *p* ≤ 0.01.

**Figure 5 pathogens-10-00286-f005:**
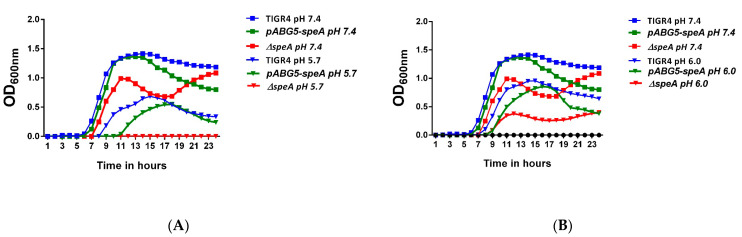
Growth of *S. pneumoniae* TIGR4, Δ*speA*, and *pABG5-speA* cultured in Todd–Hewitt broth with 0.5% yeast extract (THY) at varying pH was monitored by measuring absorbance at 600 nm and is shown in blue, red, and green, respectively, while the blank is shown in black circles in graph B. Graph (**A**) shows growth of the strains at pH 7.4 (squares) and 5.7 (triangles), and (**B**) shows growth of the strains at pH 7.4 (squares) and 6.0 (triangles). The results represent an average of three independent experiments. Statistical significance was determined by Student’s *t*-test at a significance level of *p* ≤ 0.05 and *p* ≤ 0.01.

**Table 1 pathogens-10-00286-t001:** Comparison of changes in gene expression between Δ*speA* and TIGR4 when cultured in the presence or absence of 2.5 mM hydrogen peroxide.

Gene	Description	*ΔspeA*/TIGR4 Fold Change Untreated	*ΔspeA*/TIGR4 Fold Change 2.5 mM H_2_O_2_	Function
*noxA*	Nicotinamide adenine dinucleotide phosphate oxidase	−4.6	3.6	Scavenger
*tpxD*	Thiol peroxidase	bt	−2.0	Scavenger
*psaA*	Manganese ATP-binding cassette transportersubstrate-binding lipoprotein	bt	4.1	Mn transport
*fhuD*	Iron-compound ATP-binding cassette transporter	bt	2.4	Iron transport
*ciaH*	Sensor histidine kinase	−3.0	bt	Regulator
*prtA*	Serine protease, subtilase family	3.4	3.5	Protein repair
*htrA*	High-temperature requirement A	−2.6	−2.6	Protein repair
*potD*	Putrescine ATP-binding cassette transporter	2.0	bt	Polyamine transport
*nspC*	Carboxynorspermidine decarboxylase	−2.7	−8.1	Polyamine synthesis
*aguA*	Putative agmatine deiminase	bt	−8.4	Polyamine synthesis
*speE*	Spermidine synthase	−58.7	−24.0	Polyamine synthesis

bt: below threshold; *p* ≤ 0.05 and fold change ≥ 2.

## Data Availability

Data is contained within the article or supplementary material.

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
