# Peer review of "Arginine Decarboxylase Is Essential for Pneumococcal Stress Responses"

_pathogens, 2021, doi:10.3390/pathogens10030286_

Round 1

Reviewer 1 Report

In this work, Nakamya et. al., study the role of polyamine pathway enzyme arginine decarboxylase in pneumococcal stress response. The authors utilize a classical gene deletion model in S. pneumoniae along with the complemented strain for speA. Using a variety of in vitro stress models, they demonstrate the function of speA is critical for protection against oxidative and nitrosative stress. Further, speA is found to regulate the redox balance in the cell. The experiments performed with the speA mutant and complement strains are convincing to propose the role of polyamine pathway enzyme in combating stress response. The major limitation of the study is the lack of molecular understanding of the dysregulated polyamines upon SpeA deletion and their functional role in survival of bacteria under stress. The authors have not provided all data and missed some experimental data. Overall, the study is interesting.

Major comments:

  1. The authors should include the data for GSH/GSSG-Glo assay.
  2. The authors should include the data for thermal stress response as part of the supplementary data.
  3. The statistics should be performed for Figure 4 and 5.
  4. The authors should perform One-way ANOVA when comparing more than two groups instead of Student’s t-test. The statistics needs to be revised for Fig1-3.
  5. The gene expression data will look better if represented as a heatmap instead of table. The authors if presenting the data in a table should also write the standard deviation or standard error of mean, whichever used in a separate column.
  6. Is there a reason why the complemented strain is not used for gene expression analysis and intracellular pH experiment in table 1 and Fig4, respectively?
  7. The authors should include a discussion on potential metabolic perturbations and their role in mediating survival to stress response based on their earlier manuscript and the data presented in this article.

Minor comments:

  1. For figure1 and 2, the authors should either put the data as 1 graph or split the bar graph in two separate graphs.
  2. The representation will look better if sequence of data on the graphs is changed. The data for mutant strain should come before the complemented strain data to highlight the rescue effect.

Author Response

Thanks

Reviewer 2 Report

The article by Nakamya et al. expands on previous research done by the lab on the importance of specific polyamines and the effects they have on survival. The current study focuses on the importance of the gene speA on general stress response in Streptococcus pneumoniae. The authors mostly focus on the role speA may have in oxidative stress, based on previous omics studies, but also test other stressors. The authors demonstrate that the absence of speA has a deleterious effect on the pneumococcus when exposed to oxidative stress, by either hydrogen peroxide, potassium tellurite, or S-Nitrosoglutathione. Furthermore, qRT-PCR was performed on numerous genes to examine genes effected by oxidative stress when speA is absent. Further evidence for speA utility in stress response is growth curves showing reduced growth when at lower pH. Overall the paper is well written and the experiments performed are appropriate to answer the stated question and provide a better general understanding of the role of speA in the pneumococcus. There are very few grammatical issues that should be addressed and a little more explanation needed (outlined below), but no major issues with the article as presented.

For future studies a possible mechanism for increased susceptibility to oxidative stress would be of great interest. The alteration in  gene expression patterns are a great start to getting a more detailed understanding of this event and a thorough understanding of this process would be a great contribution to the literature.

Ln 19. Should have a comma after “pathway”

Ln 84-85. Add “to” after “compared”

Ln 101-104. You tested H2O2 production and gave a value. Indicate that data is not shown.

Ln 306-307. The 5% yeast extract used in THY, is this correct? Typical THY is 0.5%, maybe up to 2%.

Figure 5. An interesting feature of the growth curves for the speA mutant is the trough you see after about hour 12, in both pH conditions. A quick sentence to speculate on this would be useful. Gene expression changes after 12 hours seem less likely, but could be a spontaneous mutant that compensates for the speA deletion. Could help identify direct interactors with speA and would be interesting to note.

Author Response

Thanks

Round 2

Reviewer 1 Report

The authors have addressed all the concerns raised by me during the first round of revision.